# Research on the Relationship between the Environmental Characteristics of Pocket Parks and Young People's Perception of the Restorative Effects—A Case Study Based on Chongqing City, China

**Huiyun Peng [1,*], Xiangjin Li [1], Tingting Yang [1] and Shaohua Tan [2]**

[1]  School of Civil Engineering and Architecture, Southwest University of Science and Technology, Mianyang 621010, China
[2]  School of Architecture and Urban Planning, Chongqing University, Chongqing 400044, China
*  Correspondence: penghuiyun@swust.edu.cn; Tel.: +86-155-0806-6160

**Abstract:** Work and life stresses can cause spiritual fatigue and emotional tension, threatening the physical and psychological health of young people. Several studies have demonstrated the important role and value of pocket parks in the emotional and spiritual refreshment of people. This study quantitatively evaluated the perceived restorative effects associated with the environmental characteristics of a pocket park, and determined the relationships between the physical-environmental characteristics, psychological-environmental characteristics and the restorative effects. In this study, pocket parks in Chongqing City were chosen as the study areas, and a total of 25 sample pictures of the parks were chosen for analysis. Each picture was quantized into 14 physical-environmental indices and three psychological-environmental indices for measurement of the restorative effect. The results showed that the environmental characteristics of parks with a restorative effect include naturality, sense of distance, charm and privacy. The physical-environmental characteristics related to young people's restoration and their degree of influence were determined through quantitative analysis. Moreover, a prediction model of the environmental restoration effect of pocket parks was established. The research conclusions can provide a reference for the evaluation and comparison of the environmental restoration performance of pocket parks and the design of restorative pocket parks.

**Keywords:** pocket park; restorative environment; stress; young people; mental health; landscape architecture

## 1. Introduction

Green spaces and the natural environment in modern cities continue to decrease, due to the increasingly fast pace of modern life, the increasing pressure of social competition and the lack of physical activities (e.g., sitting more and exercising less), especially in cities characterised by high-density development. This is an important contributing factor to mental disorders such as insomnia and depression, among others [1,2]. Mental disorders are a great challenge in modern society, and a major contributor to the global burden of disease, accounting for about 7.4% [3] of the global disease burden. According to the World Health Organization (WHO), in relation to the top ten causes of mortality globally, five of the top ten causes of disability worldwide are mental health problems [4]. Mental health issues among certain population groups, especially young people, have become a major problem [5]. Although recovery from mental disorders is related to many factors, an increasing number of studies have demonstrated that positive environmental characteristics can bring about mental and physical restoration experiences to the people living in the environment [6], including the relief of mental stress, decreased mental fatigue and unhealthy emotions, and recovery from mental health issues. The young people in

modern cities are in urgent need of restorative environments to relieve mental stress and reduce the burden of mental disorders [7,8].

The "restorative environment" theory argues that the natural environment can significantly relieve people's mental stresses, and that the separation between the population and natural environment that has been brought about by the modern urban lifestyle has a significant influence on the physical and psychological health of people. In recent years, there have been many studies examining the influence of the natural environment on the physical and psychological health of people. The psychoevolutionary model proposed by Ulrich, the stress-reduction theory and Kaplan's attention restoration theory describe, according to different perspectives, the role and mechanisms underlying the effects of the natural environment in relieving mental stress. These theories all argue that the natural environment can significantly relieve the mental stress of people living in it [9–11]. Green land in urban parks, especially green land in urban parks near residential areas, has become an important place for people to relieve their mental stress [12,13]. However, planning and construction of large-scale green parkland in cities are restricted by various factors (e.g., land shortage) against the background of high-density urban development. Pocket parks, which are scattered throughout cities, can help meet the needs for green space. Pocket parks play an important role in the natural environment. As one of the most accessible places for urban residents to become close to nature, pocket parks provide an opportunity for people to connect with nature and receive its restorative effects [14,15]. Meanwhile, as the COVID-19 pandemic continues to spread, the mental health problems of the population have become more prominent [16], which has exacerbated an enduring problem of large urban populations lacking accessible green space to fulfill the essential physical and mental health needs [17,18]. In this context, the value of pocket parks has been re-examined [19]. Pocket parks are often scattered across a city, and are considered public open space that is free for people to use. They are small in area, highly accessible and are frequently utilised, making them a good supplement to urban green spaces, and allowing people to meet their need to connect with the natural environment [15]. People have begun to realise that pocket parks are conducive to relieving mental stress and generating good restorative effects [20,21].

Relevant studies show that the environmental features of pocket parks (e.g., natural landscape, facilities and environmental perceptions) can be divided into four dimensions, namely, the naturality factor, perceptibility factor, relaxation factor and activity factor [22,23]. The naturality factor (e.g., lawn, trees, shrubs and water) is an important factor in the ability of parks to generate a restorative effect [24–26] and the value of the restorative effect is determined, to some extent, by the type, quantity and layout of the natural landscape in the park [27]. Natural sounds (e.g., water sounds, bird song and white noise generated when the wind blows leaves) which are produced by the naturality factor can block out noise and relieve stress [28–30]. The perceptibility factor refers to the subjective feelings of people in the park environment. Some studies have found that pocket parks affect people's feelings through multi-sensory perceptual stimulation including non-visual stimuli (touch, smell, hearing) and visual stimuli [31]. When the environment is considered private, quiet, safe and clean, and people can rest, read, enjoy plants and engage in other refreshing activities in the environment, people can have a better restorative experience [32]. The relaxation factor refers to the quantity of relaxation facilities and their comfort and orientation in a park. Uncomfortable chairs with no shade decrease the duration and frequency of people's visits to a park [33]. Chairs oriented towards the natural landscape are more beneficial for relieving mental stress [32]. The activity factor refers to the configuration of recreational fitness facilities and activity facilities in the park, and the quantity of activity facilities is positively related to people's engagement in physical activities in a park [34]. However, the existing studies mainly focus on the restorative value of the overall environment or some elements of pocket parks for the population, and most of them compare the environmental differences at the macro level; therefore, the quantitative evaluation of the environmental components of pocket parks is insufficient.

According to the four dimensions of the environmental characteristics of pocket parks, this study further generalizes them into physical-environment characteristics and psychological-environment characteristics. The physical environment impacts on the behavioural psychology of people, while the psychological environment is the internal driving force that promotes the construction of the physical environment. Hence, physical-environmental characteristics and psychological-environmental characteristics both influence the restorative experiences of people. Furthermore, psychological-environmental characteristics are influenced by physical-environmental characteristics. To this end, the current study aimed to quantitatively evaluate the perceived restorative effects of the pocket-park environmental characteristics on young people, and to determine the relationships between the physical- and psychological-environmental characteristics and the restorative effect of pocket parks, thus realizing the quantitative study of non-quantitative factors.

## 2. Methods

### 2.1. Sampling

There is a lack of a recognised definition of pocket parks, and thus the definition varies among research studies as a function of the research objective and background. In this study, a pocket park is defined as a public area with explicit boundaries and an open and shared centralised green space with certain activity facilities. Pocket parks have obvious natural-environmental attributes, and the area of a pocket park is generally no larger than 3000 m$^2$.

In the present study, pocket parks in the downtown area of Chongqing were chosen as the study objects. Chongqing is a mountain city. This unique geographical environment leads to dense urban population and cramped construction land, which has created obvious characteristics of high-density development. At the same time, Chongqing belongs to the typical subtropical-monsoon, humid climate, with a high-quality natural-landscape background and abundant types of pocket parks with distinctive characteristics. A total of 325 pictures of these parks were taken in the horizontal orientation. All pictures had equal dimensions and, as much as possible, most parts of the park were captured in each picture. All pictures were taken under similar weather conditions in late September, with good weather conditions, clear or slightly cloudy.

In order to achieve better test results, these pictures need to be selected and the following types of pictures should be excluded: (1) those in which the environmental perception was affected, due to the presence of many tourists; (2) those in which the environmental subjects in the park were ignored, due to an unreasonable layout; (3) those that had extremely similar compositions; (4) those in which there was inharmonious landscape in the picture, such as obvious garbage, severely damaged facilities or broken tree branches. Finally, 25 pictures were selected as the samples for further testing. Each of these 25 pictures represents a park with different characteristics.

### 2.2. Measurement of Physical-Environmental Characteristics

Physical-environmental characteristics formed a key factor in this quantitative study. In studies such as this, each factor must be refined and the indices that measure the factor or characteristic should be carefully selected, in order to comprehensively represent the factor. Based on relevant literature and previous research results, 13 indices were included in this study after comparison, integration and deletion of possible indices (Table 1). The naturality factor has the greatest influence on the restorative effect of a park. Hence, in this study, the "naturality factor" had the most comprehensive and detailed indices and was the key characteristic for quantitative evaluation. For the measurement of the "naturality factor", "hard-pavement area" and "surrounding-element area" were included as reverse-scored factors.

**Table 1.** Measurement of the physical environmental characteristics.

| Factor | No. | Indices | Interpretation | Measurement Method |
|---|---|---|---|---|
| Naturality factor | 1 | Tree and shrub area | Area of trees and shrubs in the park | The number of squares occupied by trees in the park picture was calculated using the square method. |
| | 2 | Lawn area | Area of lawns in the park | The number of squares occupied by lawns in the park picture was calculated using the square method. |
| | 3 | Water area | Area of water in the park | The number of squares occupied by water in the park picture was calculated using the square method. |
| | 4 | Plant species | Number of plant species in the park | The total number of plant species, including trees, shrubs and herbs, in the picture. |
| | 5 | Subject colour | Obvious colours and number of colours in the park | Extracted after Photoshop mosaic filtering. |
| | 6 | Terrain | Undulation on the surface | The terrain was divided into flat and uneven forms, according to the picture. |
| | 7 | Green ratio | Proportion of green plants in the field of view | The percentage of squares occupied by plants in the park picture was calculated using the square method. |
| | 8 | Hard-pavement area | Area occupied by hard pavement in the park | The number of squares occupied by hard pavement in the park picture was calculated using the square method. |
| | 9 | Surrounding-element area | Area occupied by physical elements, including buildings and vehicles outside of the park | The number of squares occupied by buildings, vehicles and other physical elements outside of the park in the park picture was calculated using the square method. |
| Relaxation factor | 11 | Area of relaxation facilities | Area of relaxation facilities in the park | The number of squares occupied by relaxation facilities in the park picture was calculated using the square method. |
| | 12 | Texture of relaxation facilities | Texture of chairs for resting in the park | Texture of all relaxation facilities in the picture. |
| Activity factor | 13 | Area of activity places | Area of activity places in the park | The number of squares occupied by activity places in the park picture was calculated using the square method. |
| | 14 | Area of entertainment and fitness facilities | Area of entertainment and fitness facilities in the park | The number of squares occupied by entertainment and fitness facilities in the park picture was calculated using the square method. |

The physical environment of the chosen pocket park was quantized by the picture-square measurement method. With respect to the quantification of pocket parks, many foreign scholars, including Shafer and Nordh, have adopted the picture-square measurement method, and have established models of environmental elements and psychological perceptions of people [25,35]. The pictures were processed by Photoshop. Each sample picture was covered completely with a 30 × 40 transparent square-grid network and different colour lumps were used to represent the corresponding evaluation indices (Figure 1). The number of squares occupied by each index was calculated, and the percentage of squares in the whole picture was computed.

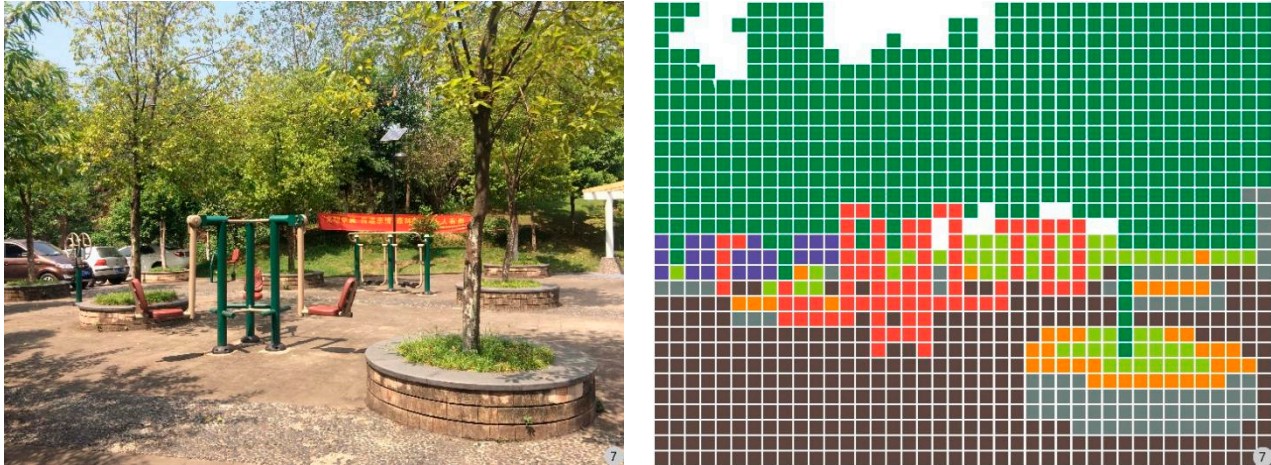

**Figure 1.** Schematic diagram of quantization method.

### 2.3. Measurement of the Psychological-Environmental Characteristics

The psychological-environmental characteristics of a park include the space atmosphere, space privacy, space safety, degree of environmental health and the surrounding-environmental isolation. Given the constraints of this study (picture stimuli, environmental sampling and grading program), space atmosphere and space safety were difficult to evaluate in the pictures, while the selection of picture samples assured the perception of environmental health. Hence, only two indices, space privacy and surrounding environmental isolation, were retained for analysis. The four features of restorative environment (being away, extent, fascination and compatibility) summarized by Kaplan, are comprehensive reflections of the psychological-environmental characteristics. Nevertheless, according to the method used by Nordh, Huttig, Hagel and Shafer, current research focuses on "being away" and "fascination", as we have more substantially constrained variation in the extent of and compatibility with our environmental sampling and rating procedures [36].

The descriptions of "environmental isolation" and "being away" were overlapped and integrated. Thus, three indices, namely, being away, fascination and privacy, were retained for analysis. These three indices were scored based on the sample pictures. Each index was described in one sentence. "Being away" was described as "this is a place where I can relax away from my busy work and learning schedule and get away from life's troubles". "Fascination" was described as "this place is charming and attracts my attention". "Privacy" was described as "I can be free from disturbances in this place".

### 2.4. Measurement of the Restorative Effect

Restorative-effect evaluation involved two factors: the restorative effect and environmental preferences. Each evaluation factor was described by one sentence. The restorative effect was described as "I can relax and engage in entertainment activities, and I can relieve my pressure and fatigue in this place". Environmental preference was described as "I like this place".

### 2.5. Picture Evaluation

In this study, college students in Chongqing who are in periods of examination or job hunting were selected as the participants, and the results of the oral interviews [37] and perceived-stress-scale tests [38] before the experiment showed that they were all in a high state of stress. The experiment was conducted in a quiet conference room on campus. Before the formal evaluation, we provided participants with a brief introduction to the research. After that, the 5 K high-definition display was used to project and play the sample photos, and five pictures similar to the evaluation pictures were presented quickly, and were used as the reference benchmark to help the participants gain a conceptual understanding of

the task of park evaluation and how to score each image. This allowed the participants to plan their scoring scale in advance. In the formal experiment, the sample pictures were presented for 10 s each. The participants recorded their response to each picture on an evaluation sheet, according to the order of presentation. All 25 sample pictures were evaluated and scored from 0 (completely disagree) to 10 (completely agree). Finally, 60 complete response sets were collected from these participants.

### 2.6. Statistical Analysis

Data in this study were analysed with correlation- and regression-analysis methods, through which the relationships between the psychological-environmental characteristics of the pocket park, the physical-environmental characteristics of the pocket park and its restorative effect were evaluated. Moreover, an environmental-restoration model of pocket parks was established. When establishing the model, the evaluation scores for the restorative effect and environmental preferences corresponding to each sample picture were standardised using the scenic-beauty-standardisation formula in the scenic-beauty evaluation (SBE). The means of all standardised values for each sample were used to calculate the standardised Z-value of the sample, which reflects the restorative effect and environmental preference of the park. The standardised Z-value reflects the restorative quality of all samples and the aesthetic evaluation of the participants. Correlation analysis and analysis of variance (ANOVA) were performed based on the standardized-restorative-effect and environmental-preference scores. Later, multiple-stepwise-linear-regression analysis was carried out using SPSS 22.0, and a relation model between the restorative effect and evaluation indices was established.

$$Z_{ij} = (R_{ij} - \overline{R}_j)/S_j$$
$$Z_i = \sum_j Z_{ij}/N_j$$

where $Z_{ij}$ is the standardised value of participant $j$ to sample picture $i$, $R_{ij}$ is the score of participant $j$ for sample picture $i$, $\overline{R}_j$ is the mean score of participant $j$ for all sample pictures, $S_j$ is the standard deviation of the scores of participant $j$ for all sample pictures, $Z_i$ is the standardised score of sample picture $i$, and $N_j$ is the total number of participants.

## 3. Results

### 3.1. Comparison of the Standard Scores for the Restorative Effect and Environmental Preference

Comparison of the standard scores for the restorative effect and environmental preference (Figure 2) revealed that the trend in variation in the restorative effect was generally consistent with that of the environmental preference, indicating that young people prefer an environment with a high restorative effect. In other words, environments that young people like also have a good restorative effect. A regression analysis was then performed, using SPSS 22.0. The results (Table 2) showed that the restorative effect significantly influenced the environmental preference, with a positive correlation between the two. These results are consistent with the conclusion of Kaplan that "an environment that people prefer is very likely to be a restorative environment".

**Table 2.** Regression coefficients of models [a].

| Model | | Non-Standardised Coefficient | | Standardised Coefficient | T | Sig. |
|---|---|---|---|---|---|---|
| | | B | Standard Deviation | Beta | | |
| 1 | (Constant) | $-3.824 \times 10^{-11}$ | 0.042 | | 0.000 | 1.000 |
| | Restorative effect | 0.956 | 0.048 | 0.972 | 19.777 | 0.000 |

[a]. Dependent variable: Environmental preferences.

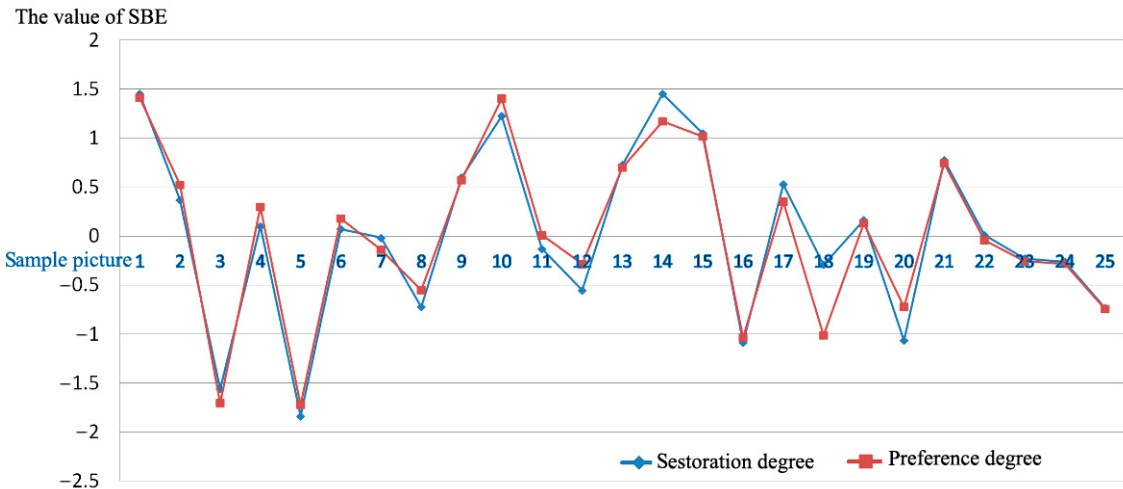

**Figure 2.** Comparison of restorative-effect and environmental-preference scores.

*3.2. Correlation Analysis between Physical- and Psychological-Environmental Characteristics*

When examining the evaluation results of the three psychological-environmental characteristics (Figure 3), there appeared to be differences in the evaluation results between almost all samples, except for a few samples (Sample 3, Sample 5 and Sample 25). Hence, the evaluation results for each characteristic had to be analysed independently, to determine which physical-environmental characteristics had positive effects and which had negative effects on the evaluation results.

(1)     Effects of the physical-environmental characteristics of the pocket park on "being away"

According to the multiple-linear-regression results (Table 3), "being away" was influenced by the surrounding element area, plant species and green ratio, with *p* values of 0.001, 0.002 and 0.016, respectively. This indicates that there were significant linear relations between these variables.

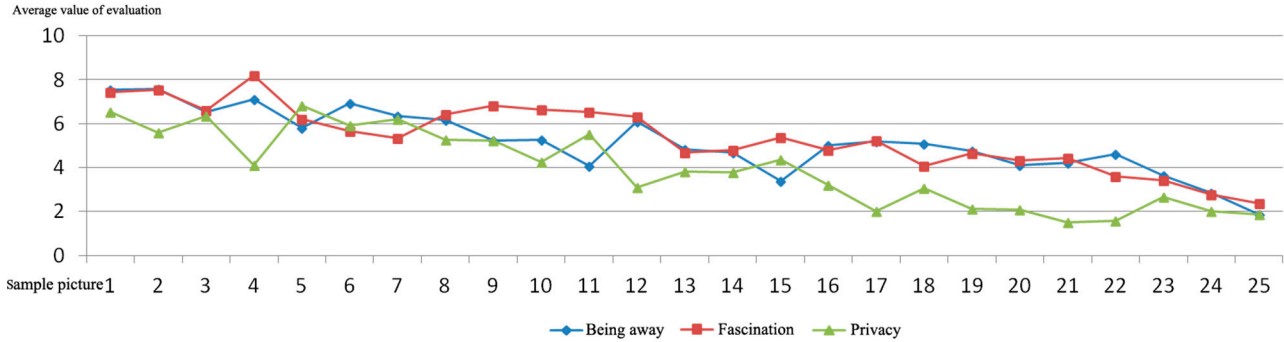

**Figure 3.** Evaluation results of the psychological-environmental characteristics.

The influence of the surrounding-element area on "being away" was the greatest, with a B value of −0.508, indicating that there was a negative relationship between the surrounding-element area and "being away". In other words, a larger surrounding-element area was associated with a lower evaluation score of being away. Plant species was positively correlated with being away (B value = 0.403), indicating that the presence of more plant species was associated with a higher evaluation score of being away. There was also a positive correlation between the green ratio and being away (B value = 0.335), indicating that the evaluation score of being away increased with an increasing green ratio.

**Table 3.** Multiple-regression analysis between "being away" [a] and the physical environment of the pocket park.

| Models | | Non-Standardised Coefficient | | Standardised Coefficient | T | Sig. |
|---|---|---|---|---|---|---|
| | | B | Standard Deviation | Beta | | |
| 1 | (Constant) | $-2.751 \times 10^{-16}$ | 0.149 | | 0.000 | 1.000 |
| | Surrounding-element area | −0.687 | 0.152 | −0.687 | −4.531 | 0.000 |
| 2 | (Constant) | $-5.250 \times 10^{-16}$ | 0.128 | | 0.000 | 1.000 |
| | Surrounding-element area | −0.650 | 0.131 | −0.650 | −4.957 | 0.000 |
| | Plant species | 0.393 | 0.131 | 0.393 | 2.994 | 0.007 |
| 3 | (Constant) | $-5.809 \times 10^{-16}$ | 0.114 | | 0.000 | 1.000 |
| | Surrounding-element area | −0.508 | 0.129 | −0.508 | −3.939 | 0.001 |
| | Plant species | 0.403 | 0.117 | 0.403 | 3.451 | 0.002 |
| | Green ratio | 0.335 | 0.128 | 0.335 | 2.609 | 0.016 |

[a]. Dependent variable: being away.

(2)  Effects of the physical-environmental characteristics of the pocket park on fascination

According to the multiple-linear-regression-analysis results (Table 4), "fascination" was influenced by the green ratio, water area and plant species. The *p* values were 0.001, 0.004 and 0.050, respectively.

**Table 4.** Multiple-linear-regression analysis of "fascination" [a] and the physical environment of the pocket park.

| Models | | Non-Standardised Coefficient | | Standardised Coefficient | T | Sig. |
|---|---|---|---|---|---|---|
| | | B | Standard Deviation | Beta | | |
| 1 | (Constant) | $-6.185 \times 10^{-16}$ | 0.170 | | 0.000 | 1.000 |
| | Plant species | 0.554 | 0.174 | 0.554 | 3.194 | 0.004 |
| 2 | (Constant) | $-6.743 \times 10^{-16}$ | 0.151 | | 0.000 | 1.000 |
| | Plant species | 0.551 | 0.154 | 0.551 | 3.564 | 0.002 |
| | Green ratio | 0.410 | 0.154 | 0.410 | 2.652 | 0.015 |
| 3 | (Constant) | $-5.983 \times 10^{-16}$ | 0.126 | | 0.000 | 1.000 |
| | Plant species | 0.310 | 0.149 | 0.310 | 2.084 | 0.050 |
| | Green ratio | 0.546 | 0.136 | 0.546 | 4.027 | 0.001 |
| | Water area | 0.502 | 0.154 | 0.502 | 3.249 | 0.004 |

[a]. Dependent variable: fascination.

The green ratio had the greatest influence on "fascination". The B value was 0.546, indicating a positive correlation between the green ratio and "fascination". In other words, the evaluation score of fascination was higher when the green ratio was higher. Water area had the next strongest influence on fascination, with a positive correlation between the two variables (B value = 0.502), indicating that the evaluation score was higher when there was a larger water area. "Fascination" was also positively related with plant species (B value = 0.335), such that the evaluation score of fascination increased with increasing plant species.

(3)  Effects of the physical-environmental characteristics of the pocket park on "privacy"

According to the multiple-linear-regression results (Table 5), "privacy" was influenced by tree and shrub area, area of activity places and plant species. The *p* values were 0.010, 0.018 and 0.036, respectively.

**Table 5.** Multiple-linear-regression analysis of "privacy" [a] and the physical environment of the pocket park.

| Models | | Non-Standardised Coefficient | | Standardised Coefficient | T | Sig. |
|---|---|---|---|---|---|---|
| | | B | Standard Deviation | Beta | | |
| 1 | (Constant) | $-1.998 \times 10^{-16}$ | 0.164 | | 0.000 | 1.000 |
| | Tree and shrub area | 0.599 | 0.167 | 0.599 | 3.587 | 0.002 |
| 2 | (Constant) | $-1.366 \times 10^{-16}$ | 0.142 | | 0.000 | 1.000 |
| | Tree and shrub area | 0.463 | 0.152 | 0.463 | 3.040 | 0.006 |
| | Area of activity places | −0.444 | 0.152 | −0.444 | −2.912 | 0.008 |
| 3 | (Constant) | $-3.449 \times 10^{-16}$ | 0.131 | | 0.000 | 1.000 |
| | Tree and shrub area | 0.404 | 0.143 | 0.404 | 2.834 | 0.010 |
| | Area of activity places | −0.371 | 0.144 | −0.371 | −2.578 | 0.018 |
| | Plant species | 0.317 | 0.142 | 0.317 | 2.240 | 0.036 |

[a]. Dependent variable: privacy.

Tree and shrub area was the strongest influential factor on "privacy" with a B value of 0.404, indicating a positive relation between these two factors. The evaluation score of privacy was higher when the tree and shrub area was larger. The area of activity places was negatively related to the evaluation score of privacy (B value = −0.371), indicating that the evaluation score decreased with increasing area of activity places. Plant species was positively related to privacy (B value = 0.317), indicating that the evaluation score of privacy was higher when there were more plant species.

### 3.3. Correlation Analysis between the Physical-Environmental Characteristics and the Restorative Effect

The Pearson correlation analysis was performed, using SPSS 22.0 to evaluate the correlation between each physical-environmental characteristic and the restorative effect. The results are shown in Table 6. The strength of the relationships between the restorative effect and each characteristic was as follows: green ratio > plant species > tree and shrub area > area of activity places > water area > surrounding element area. Specifically, tree and shrub area, water area, plant species and green ratio were all positively related to the restorative effect, while the surrounding-element area and area of activity places were negatively correlated.

Based on the analysis, it can be concluded that the restorative effect is a collaborative consequence of several physical-environmental characteristics, and there were autocorrelations among the variables. For instance, the correlation coefficient between "tree and shrub area" and "green ratio" reached as high as 0.790, and the correlation coefficient between "area of activity places" and "hard-pavement area" was 0.631, indicating that there might be collinearity problems between the variables. Therefore, it was necessary to analyse the collaborative effect of these factors. Multiple-linear-regression analysis can reveal the relationships between several independent variables and dependent variables. To avoid collinearity problems, a multiple-linear-regression analysis was adopted to establish a restorative-evaluation model.

### 3.4. Physical-Environmental Restorative-Evaluation Model of the Pocket Park

A multiple-linear-stepwise regression was performed using SPSS 22.0 Chicago, IL, USA, with the evaluation score of the restorative effect as the dependent variable and the 13 chosen physical-environmental characteristics as the independent variables. Through the multiple-linear-stepwise regression, factors that were the least important and autocorrelated factors were gradually eliminated. Finally, three factors (green ratio, water area and plant species) were retained to establish an environmental restorative-evaluation

model. The results are shown in terms of the model summary (Table 7), ANOVA (Table 8) and model regression coefficients (Table 9).

**Table 6.** Pearson correlation analysis.

| Physical-Environmental Characteristics | | Being Away |
|---|---|---|
| Pearson Correlation | Tree and shrub area | 0.472 |
| | Lawn area | 0.107 |
| | Water area | 0.384 |
| | Plant species | 0.514 |
| | Number of subject colours | 0.187 |
| | Terrain | 0.264 |
| | Green ratio | 0.515 |
| | Hard-pavement area | −0.328 |
| | Surrounding-element area | −0.374 |
| | Area of relaxation facilities | 0.169 |
| | Texture of relaxation facilities | −0.129 |
| | Area of activity places | −0.461 |
| | Area of entertainment and fitness facilities | −0.070 |
| Significance (<0.05 is significant) | Tree and shrub area | 0.009 |
| | Lawn area | 0.306 |
| | Water area | 0.029 |
| | Plant species | 0.004 |
| | Number of subject colours | 0.185 |
| | Terrain | 0.101 |
| | Green ratio | 0.004 |
| | Hard-pavement area | 0.055 |
| | Surrounding-element area | 0.033 |
| | Area of relaxation facilities | 0.210 |
| | Texture of relaxation facilities | 0.269 |
| | Area of activity places | 0.010 |
| | Area of entertainment and fitness facilities | 0.369 |

**Table 7.** Model summary.

| Model | Multiple Correlation Coefficient | Coefficient of Determination | Adjusting the Coefficient of Determination | Standard Deviation | Durbin–Watson |
|---|---|---|---|---|---|
| 1 | 0.515 [a] | 0.265 | 0.233 | 0.87582574 | |
| 2 | 0.747 [b] | 0.558 | 0.518 | 0.69403444 | 1.601 |
| 3 | 0.797 [c] | 0.635 | 0.582 | 0.64624846 | |

[a]. Predicted value: (Constant), green ratio; [b]. Predicted value: (Constant), green ratio and water area; [c]. Predicted value: (Constant), green ratio, water area and plant species.

**Table 8.** ANOVA.

| | Models | Sum of Squares | df | Mean Square | F | Sig. |
|---|---|---|---|---|---|---|
| | Regression | 15.230 | 3 | 5.077 | 12.155 | 0.000 [a] |
| | Residual error | 8.770 | 21 | 0.418 | | |
| 3 | Total | 24.000 | 24 | | | |

[a]. Dependent variable: evaluation score of restorative effect.

It can be seen from the model summary (Table 7) that model 3 had the maximum coefficient of determination and the minimum standard deviation after adjustment. The multiple correlation coefficient (R), coefficient of determination and the adjusted coefficient of determination were 0.797, 0.635 and 0.582, respectively. This indicates that the goodness of fit was relatively high, and there were no unexplained variables. The Durbin–Watson (DW)-test statistic was applied to test whether there were auto-correlations in the model.

Generally, a DW value between 1.5 and 2.5 indicates no auto-correlations. In this study, the DW value was 1.601, which indicates an absence of serious collinear relations among the independent variables.

**Table 9.** Regression coefficients of models [a].

| Models | | Non-Standardised Coefficient | | Standardised Coefficient | T | Sig. |
|---|---|---|---|---|---|---|
| | | B | Standard Deviation | Beta | | |
| 1 | (Constant) | $-2.873 \times 10^{-17}$ | 0.175 | | 0.000 | 1.000 |
| | Green ratio | 0.515 | 0.179 | 0.515 | 2.879 | 0.008 |
| 2 | (Constant) | $-1.145 \times 10^{-16}$ | 0.139 | | 0.000 | 1.000 |
| | Green ratio | 0.665 | 0.147 | 0.665 | 4.525 | 0.000 |
| | water area | 0.562 | 0.147 | 0.562 | 3.825 | 0.001 |
| 3 | (Constant) | $-2.910 \times 10^{-16}$ | 0.129 | | 0.000 | 1.000 |
| | Green ratio | 0.618 | 0.139 | 0.618 | 4.457 | 0.000 |
| | Water area | 0.398 | 0.158 | 0.398 | 2.521 | 0.020 |
| | Plant species | 0.318 | 0.152 | 0.318 | 2.091 | 0.049 |

[a]. Dependent variable: evaluation score of restorative effect.

It can be seen from the ANOVA results (Table 8) that the F-tests were significant at 0.000 < 0.05, indicating that all three characteristics in the model were significantly correlated with the restorative effect. Therefore, a linear model could be established.

It can be seen from the regression coefficients in Table 9 that the standardised coefficient ranged between −1 and 1. A higher absolute value of the standardised coefficient implies a higher influence of the predictive variable on the dependent variable and a greater explanatory effect of the dependent variable. The *p* values were 0.000, 0.020 and 0.049, indicating significant influences at the *p* = 0.05 level and significant linear relations. Hence, the non-standardised regression equation was gained from the regression-coefficient table: restorative degree = $-2.910 \times 10^{-16}$ + 0.618 × green ratio + 0.398 × water area + 0.318 × plant species.

Nevertheless, the non-standardised regression equation contains a constant, and it cannot compare the relative importance of predictive variables. Therefore, it is common to transform the original regression equation into a standardised regression equation based on the standardised coefficients. In other words, the prediction model for the environmental-restorative effect of a pocket park is as follows:

Restorative effect = 0.618 × green ratio + 0.398 × water area + 0.318 × plant species

This equation indicates that the restorative effect increases by 0.618 units for every unit increase in the green ratio, by 0.398 units for every unit increase in the water area, and by 0.318 units for every unit increase in the plant species in a pocket park environment. The constructed model is applicable to the evaluation and comparison of the environmental-restorative effects of pocket parks.

## 4. Discussion

According to the research results, the environmental characteristics that produce a restorative effect primarily include naturality, being away, fascination and privacy (Figure 4).

### 4.1. Naturality

The naturality of a pocket-park environment is the primary characteristic that meets young people's restorative demands. According to studies, the colour, types and quantity of a natural landscape (e.g., flowers and plants, trees, shrubs and water) are very important in the restorative environmental evaluation of the park; excessive non-natural landscapes (e.g., hard pavement and activity places) are negatively correlated with a restorative effect.

In other words, naturality plays a dominant role in the restorative effect of a pocket park. The established prediction equation of the environmental-restorative effect of pocket parks indicates that the three most important characteristics are the green ratio, water area and plant species. All of these three characteristics reflect the naturality of the park.

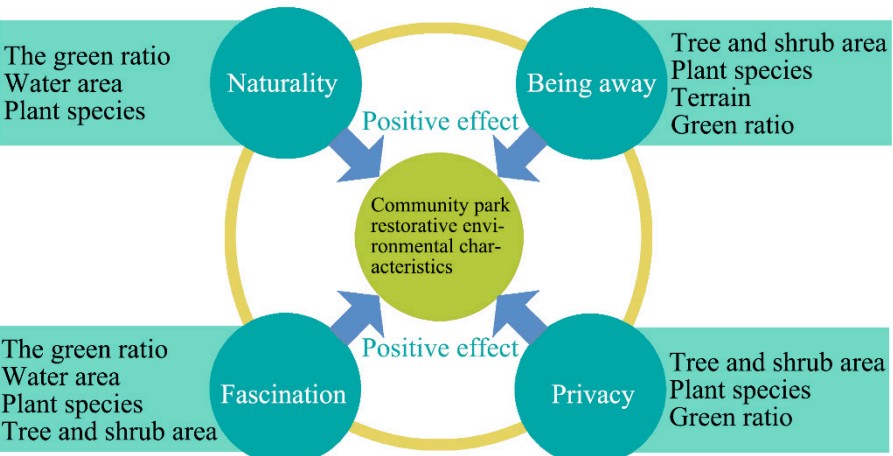

**Figure 4.** Restorative environmental characteristics of the pocket park.

The green ratio reflects the "quantity" of green space in a park and the current results indicate that the restorative effect is positively related to the green ratio. However, there was variation in the restorative effect as a function of a high or low green ratio. When the green ratio was lower than 45%, the restorative-effect-evaluation score was relatively lower, and there was a positive correlation between them. When the green ratio was higher than 45%, the restorative-effect-evaluation score was relatively higher. and the correlation between the two variables was irregular. The restorative-effect-evaluation score is more sensitive to other factors (e.g., water).

Water is a natural element that is closely related to the emotions of people, and the restorative effect generated by a natural landscape containing water is significant. Among the 25 sample pictures, there were five samples containing water, and these samples were rated in the top 10 in terms of their restorative effect. The top three samples all contained water. Although water areas are different, they have positive restorative effects.

Plant species increase the interest and landscape variation of a natural space. Life is the most prominent characteristic of plants. Plants can grow and change differently. With seasonal and growth changes, the colours, textures, leaf density and all other characteristics of plants continuously change. These changes are intensified with increased plant species, enriching the vitality of the natural environment. People who connect with environments with diversified plant species will have more opportunities to decrease their psychological fatigue and improve their attention, producing positive restorative effects.

### 4.2. Being Away

"Being away" was found to be a major characteristic of pocket parks that contributes to meeting the restorative demands of the young people; the "surrounding-element area" was found not to be the primary influencing factor. This influence was found to be negative. "Surrounding elements" refers to buildings, labels, traffic facilities and physical elements outside of the park that people can see from the park. The "being away" evaluation score was lower when the surrounding-element area was larger. Because the involvement of surrounding elements reduces people's sense of being far away in the park, it is still closely connected with things in daily life.

Plant species and the green ratio were secondary influential factors on the "being away" evaluation of the park's restorative environment. Both factors had positive impacts. The "being away" evaluation score was higher when there was an abundance of plant

species and a greater green ratio. Rich plant species bring about plant appreciation and can attract people's attention, thus generating a sense of distance. A higher green ratio implies a higher number of green plants and fewer corresponding artificial elements (e.g., hard pavement and surrounding elements). People may perceive a sense of distance when appreciating the exuberant green plants, which, in turn, relieves their mental pressure and helps provide refreshment. Meanwhile, rich plant species and large-scale green space also form a barrier to filter noise and scenes from the surrounding urban life, forming a defined park space.

### 4.3. Fascination

Fascination is another major characteristic of pocket parks that can meet restorative demands. Young people feel that a park environment with fascination is charming and not boring. Such an environment can not only attract young people's attention, but also helps them feel relaxed and refreshed. Meanwhile, it promotes people's participation in behavioural activities in the park, and encourages people to interact with the park environment, leading to restorative-health effects. The physical-environmental characteristics of a pocket park can directly influence its fascination evaluation. Specifically, the green ratio, water area, plant species and tree and shrub area had positive impacts on fascination, while the hard-pavement area and area of activity places had negative impacts on fascination.

There was a positive correlation between the green ratio and fascination, and this was the strongest influential factor. Water area was the secondary influencing factor on fascination. Static water can not only create a quiet atmosphere, but also increases the spatial hierarchy and expands space perceptions, to increase the visual integration of spaces. Dynamic water can not only reflect energy and vitality, but also brings visual and acoustic perceptions. The sound of moving water can also effectively mask surrounding noises. Combined with small landscapes, dynamic water can be viewed as a visual focus to increase the interest of a space. For example, sample pictures containing water had relatively higher evaluation scores for fascination. When water could be seen in the pictures, it attracted the attention of the observers, regardless of how much water could be seen, thus increasing the fascination evaluation score. Plant species and tree and shrub area also positively influenced fascination. Similar to the green ratio and water area, plant species and tree and shrub area also reflect the composition of the natural environment, making the natural environment more charming and attractive. When young people pay attention to the natural environment, it can help them to forget the pressures of their daily lives, thus relieving their psychological fatigue and promoting a restorative effect.

### 4.4. Privacy

The privacy of a pocket park is a major characteristic that helps meet the restorative demands of the young people. The physical-environmental characteristics of a park directly influence the evaluation of the privacy of the park. Tree and shrub area, plant species and green ratio were positive influencing factors on privacy, while hard-pavement area, area of activity places, and area of entertainment and fitness facilities were negative influencing factors.

Tree and shrub area was the primary influencing factor on privacy, with a positive correlation between tree and shrub area and privacy. Trees and shrubs are taller than lawns and vegetation. When the tree and shrub area is large, there are many plant barriers formed by the trees and shrubs, which often generate deep and serene feelings. Moreover, the enclosing and shielding effect of trees and shrubs is beneficial to creating private and semi-private spaces, allowing people to maintain a controlled space, free from external disturbances. Area of activity places was the secondary influencing factor; the evaluation score of privacy decreased with increasing area of activity places. Large activity spaces are often areas with concentrated human activities. Moreover, such places are mainly open, and it is difficult to obtain privacy there. The behavioural activities of young people in such environments are impacted by the surrounding activities and the lack of privacy.

## 5. Conclusions

Based on the research of relevant scholars, this paper is a survey and research looking at young people under pressure. This paper quantitatively evaluates the perceived restoration of the environmental characteristics of pocket parks, and establishes a prediction model for the environmental restoration effect of pocket parks. It further summarizes the characteristics of the park environment that have a restorative effect on young people, including "naturality"," being away", "fascination" and "privacy". The results will provide a theoretical basis for the future planning and design of pocket parks that are more in line with the spiritual needs of young people.

However, there were some limitations to this study. This study employed college students as an example of young people, but the young people from other cultures and occupations have different needs for restorative environments, which may have resulted in bias. In the future, we need to conduct a more comprehensive study according to the characteristics of different types of pocket parks and the differences among different users. At the same time, the quantification of the physical-environment elements of the pocket park is mainly extracted from the perspective of visual experience. Future research will further expand the dimension of quantification, and conduct in-depth quantitative analysis on the restoration experience of elements such as hearing, smell, and touch, which exist in the environment.

**Author Contributions:** Conceptualization, H.P.; Methodology, H.P.; Validation, H.P.; Formal analysis, X.L. and T.Y.; Investigation, X.L. and T.Y.; Resources, S.T.; Data curation, T.Y.; Writing—original draft, H.P.; Writing—review & editing, X.L.; Visualization, X.L.; Supervision, S.T. All authors have read and agreed to the published version of the manuscript.

**Funding:** This study was supported by the Natural Science Foundation of China (51808463, 51478057) and supported by Sichuan Science and Technology Program (2023NSFSC1051).

**Institutional Review Board Statement:** Not applicable.

**Informed Consent Statement:** Informed consent was obtained from all subjects involved in the study.

**Data Availability Statement:** The data presented in this study are available on request from the corresponding author. As questionnaire data involves the privacy of participants, the data has not been made publicly available.

**Acknowledgments:** We would like to thank the students for participating in the photo-test experiment.

**Conflicts of Interest:** The authors declare no conflict of interest.

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
