# Peer review of "Research on the Relationship between the Environmental Characteristics of Pocket Parks and Young People’s Perception of the Restorative Effects—A Case Study Based on Chongqing City, China"

_sustainability, doi:10.3390/su15053943_

Round 1
Reviewer 1 Report
1. People's demand for a better human settlements is higher and more diversified, so the research on the Relationships between the Environmental Characteristics of Pocket Parks and their Perceived Restoration Effects has high social value and practical significance.
2. The previous research starts from four dimensions(the naturality factor, perceptibility factor, relaxation factor and activity factor), and the author's research starts from three dimensions (Table 1). What is the new theoretical contribution of this article? What is the innovation of this article?
3. Data source: a total of 325 photos were taken. Where are these photos distributed in Chongqing? According to what principles are the photos selected? Which pocket parks? Where are these parks located? Many key data sources are unclear to readers.
4. As for the selection of participants, “the environmental evaluation results of students are highly consistent with those of non-students”, first of all, this conclusion is not the research conclusion of this article. Second, students are only a part of many social groups, and the data of 60 college students as a sample obviously cannot support the research conclusion of this article; The other 60 students and 325 photos from the other two universities do not match the topic of the paper, taking Chongqing as an example. It is suggested to improve it seriously.
5. In the discussion and conclusion part, the data source and material support of the article are only a small part of some social groups and Chongqing. The discussion and conclusion are not targeted and specific discussions and conclusions, but are amplified. It is suggested to conduct objective and real analysis and research based on the existing data of the article.
6. For details, the number of references is small, and the proportion of references in the last five years is very small, and there are many errors, such as 3/5/11. It is recommended to modify and improve.

Author Response
请参阅附件。

Reviewer 2 Report
The topic of this paper is well known and the arguments of the discussion haven’t originality. Anyway, the article is clear and has a strong metodological approach.
I would also tell that the main concerns of the manuscript, as already reported in my report, determine a decision of acceptance pending minor revisions. Minor revisions are mainly due to the fact that (i) the manuscript needs a better definition, academic ground and positioning in the discipline, which is now a bit rather generic and not completely addressing the main disciplinary issue, (ii) the manuscript also needs a more detailed part explaining originality and novelty of the paper, which can be addressed in the introduction or in the discussion, depending on the final decision of the author; (iii) the manuscript finally needs a broader conclusion section, especially to clarify the possible , future research lines, so we encourage authors to better details the future research that can stem from this paper. Finally, I would suggest a complete and thorough revision of the literature. In this perspective, I would suggest a better definition and description of the policy implications stemming from the empirical approach of this study. I remain sympathetic with the paper, but I would see some revisions before publication. I believe these revisions will contribute to improve the quality and readability of this text.
Reviewer 3 Report
The article presents a way to quantitatively evaluate the perceived restoration of the environmental features of pocket areas based on pixellating images of the locations. The statistical analysis is rigorous. However, there are some issues which the authors can consider to make the contribution better. First, could you tell me what evidence basis was the image sampling done in section 2.1? Have you been able to follow any secondary sources to explain the sampling process? Secondly, the conclusion section is relatively brief, and it is tough to understand the takeaways of this investigation. It can be extended to align with the abstract, introduction and the results derived from the analysis.
Round 2
Reviewer 1 Report
A total of 325 photos were taken. Where are these photos distributed in Chongqing? When was the data collected? Which pocket parks are the 25 photos distributed in? Where are these pocket parks located in Chongqing, China? How do the 25 pictures represent Chongqing, China(Title)? How are 60 students from the two universities selected? According to what principles? When will it take place? It is suggested to supplement and improve the key data sources seriously.
